# The Role of EuroSCORE II in Predicting Postoperative Pressure Injuries in Cardiac Surgery Patients: A Cross-Sectional Study

**DOI:** 10.3390/healthcare13222880

**Published:** 2025-11-12

**Authors:** Dijana Babić, Snježana Benko Meštrović, Želimir Bertić, Milan Milošević, Antonija Herceg, Ana Miloš

**Affiliations:** 1Magdalena-Clinic for Cardiovascular Diseases, Faculty of Medicine J.J. Strossmayer University of Osijek, Ljudevita Gaja 2, 49217 Krapinske Toplice, Croatia; 2Department of Nursing, Catholic University of Croatia, Ilica 244, 10000 Zagreb, Croatia; 3Center for Cardiopulmonary Rehabilitation, University Hospital Sveti Duh, 10000 Zagreb, Croatia; 4Department of Physiotherapy, University North, Ul. 104 Brigade 3, 42000 Varazdin, Croatia; 5Department of Physiotherapy, Faculty of Applied Health Studies in Rijeka, Ul. Viktora Cara Emina 5, 51000 Rijeka, Croatia; 6Institute of Public Health of Bjelovar-Bilogora County, 43000 Bjelovar, Croatia; 7Department of Nursing, Faculty of Health Studies, Ul. Viktora Cara Emina 5, 51000 Rijeka, Croatia; 8School of Medicine, Andrija Stampar School of Public Health, University of Zagreb, Rockefellerova 4, 10000 Zagreb, Croatia; 9Intensive Care Unit, Magdalena-Clinic for Cardiovascular Diseases, Faculty of Medicine J.J. Strossmayer University of Osijek, Ljudevita Gaja 2, 49217 Krapinske Toplice, Croatia; 10Department for Cardiovascular Surgery, Magdalena-Clinic for Cardiovascular Diseases, Faculty of Medicine J.J. Strossmayer University of Osijek, Ljudevita Gaja 2, 49217 Krapinske Toplice, Croatia

**Keywords:** pressure injury, risk assessment, EuroSCORE II index, cardiac surgery

## Abstract

**Highlights:**

**Abstract:**

**Background/Objectives:** Pressure injuries (PIs) are an increasing public health concern, particularly affecting hospitalised patients with limited mobility and chronic illnesses, such as those undergoing cardiac surgery. The EuroSCORE II index, a validated model for predicting operative risk in cardiac surgery, can serve as an accurate PI risk assessment tool for cardiac surgery patients. **Methods:** A cross-sectional study was conducted on patients undergoing cardiac surgery over a six-month period. The sample consisted of patients selected according to the calculated EuroSCORE II index and admitted for elective surgical procedures. The Braden Scale was used for the standard preoperative and postoperative PI risk assessment. Categorical variables are shown as frequencies with corresponding percentages. Continuous variables are presented as median and interquartile range. Group differences in continuous variables according to EuroSCORE II were analysed using the Mann–Whitney U test, with the Hodges–Lehmann estimator of the median difference and the corresponding 95% confidence interval. **Results:** The assessment showed that patients with a medium and high EuroSCORE II index (>4.0) were significantly older (M = 73; IQR: 68–77), with a higher preoperative Braden score (M = 20; IQR: 17–21), longer intraoperative total time (M = 6; IQR: 5–7) and overall longer duration of hospitalisation (M = 14; IQR: 10–21). A statistically significant difference (*p* = 0.043) was observed in the occurrence of PI after the procedure. Within the group of patients with a medium/high EuroSCORE II index, the recorded frequency of PI after the procedure was 30.8%, compared to the group of patients with a low EuroSCORE II index, among whom the observed frequency was 17.6%. **Conclusions:** The findings of this study show that a higher EuroSCORE II index is significantly associated with an increased risk and incidence of PI in patients undergoing cardiac surgery, highlighting its potential use as a predictive tool for postoperative PI risk stratification.

## 1. Introduction

The occurrence of pressure injury (PI) is a growing and serious public health problem that significantly impairs the safety and recovery of patients admitted for hospital treatment [1,2]. The population at risk of developing PI is increasing, mainly due to the rising proportion of elderly people and the consequent increase in the incidence of chronic diseases, as well as long-term impaired mobility [3].

Pressure injury is defined as localized skin and/or subcutaneous tissue damage that occurs as a result of prolonged pressure exposure or a combination of pressure, friction and shear [4,5,6]. PIs typically develop over bony prominences but can also be present after applications of medical or other devices/equipment directly on the skin [5,7]. Hospital-acquired PIs (HAPIs) occur in 6.4% to 17.4% of all hospital admissions [8,9,10,11]. The occurrence of PI in a patient during hospitalisation is associated with pain, infection, emotional distress, prolonged treatment, and even fatal outcomes [6,12]. The treatment of patients who experience PI or related complications requires additional human and material resources, resulting in increased total hospital costs [13,14,15].

The frequency of PI after cardiac surgery remains relatively high and is measured in the range of 5.8% to 29.5%, depending on the methodology of data collection and analysis of individual studies [16,17,18,19]. There are several reasons for the high incidence of PI in cardiac surgery patients, including their complex health status, which is often further complicated by various comorbidities and the effects of a prolonged operation [20]. Patients who have undergone cardiac surgery are exposed to periods of poor tissue oxygenation, hypoperfusion, prolonged immobility, use of vasopressor drugs, and fluctuations in circulating volume, which represent an additional risk for the occurrence of pressure injuries [20,21,22,23]. The prevention of pressure injuries relies on timely risk assessment, which forms the foundation for planning and implementing the necessary measures. Several standardized tools for assessing the risk of PI have been developed and continue to represent the basis of all preventive measures aimed at reducing the incidence of PI [24,25,26]. The best known and most commonly used are the Norton, Waterlow, Knoll, and Braden scales, which are mainly based on the observation of common risk factors for the occurrence of PI in a representative group of patients [24,25,27,28] However, there is no general consensus about the variables that would represent the most important risk indicators for the occurrence of PI in different patients or specific environments [25].Risk assessment scales are often criticized for their poor psychometric properties and inability to ultimately improve and optimize treatment outcomes [29,30]. Also, the terminology used in many of them is often confusing and ambiguous, resulting in misinterpretations and scoring errors [24]. Very often, the observed risk factors do not include parameters specific to acute hospital environments, such as the type of operation, stay in intensive care and the use of parenteral nutrition [24]. This information appears particularly important when assessing the risk of PI in patients undergoing cardiac surgery, as existing aids rarely consider both the specifics of the patients and the complexity of the surgical procedure, thus providing justification for revision and optimisation [18]. The difficulties in finding a standardized risk assessment tool that would be acceptable for use in specific patient groups have prompted the scientific and professional community to reconsider the use of other models that could help identify individuals at high risk of PI development.

The EuroSCORE index is one of the best validated and most commonly used predictive models for assessing preoperative risk and postoperative mortality in cardiac surgery [31,32,33,34]. The goal of designing and creating the EuroSCORE index was to determine the risk profile of adult cardiac surgery patients and the risk of mortality during and after the procedure [35,36]. It has been in use since 1999, when the original version was developed. The EuroSCORE II version was designed and released for use in 2012, on the basis of more recent clinical indicators [32,35,36,37,38]. It is now routinely used in many cardiac surgery centres worldwide [39]. Constituent parts of the EuroSCORE II index are variables related to the patient’s general condition (e.g., age, sex, chronic diseases), cardiac function factors (The New York Heart Association (NYHA) and the Canadian Cardiovascular Society (CCS) classification, ejection fraction (EF), recent myocardial infarction (MI), pulmonary hypertension), and factors related to the surgical procedure (e.g., urgency, severity of procedure and operation of thoracic aorta) [38]. The scoring system identifies three groups of risk factors (Patient-related, Operation-related and Cardiac factors) and combines these variables into a single percentage to estimate the risk of 30-day mortality from cardiac surgery. The initial version of the index defined three risk groups of patients according to the specified parameters: the low-risk group (EuroSCORE 1–2), the medium-risk group (EuroSCORE 3–5) and the high-risk group (EuroSCORE 6 plus) [37].

In recent years, the scientific community has recognised that such evaluation systems can be applied in practice to assist in choosing a treatment model and predicting perioperative complications, while also planning costs, comparing results, improving quality, and making long-term predictions [39,40]. The EuroSCORE index has been identified as a potentially useful tool for determining the level of risk for developing PI after cardiac surgery, precisely because of its ability to encompass various risk factors contributing to PI development, such as advanced age, chronic health conditions, and specific treatment circumstances. Since the incidence of pressure injuries after cardiac surgery remains relatively high compared to other surgical procedures, it seems reasonable to consider using this index to identify patients who have an increased likelihood of developing this postoperative complication.

The aim of the study is to determine the predictive value of the EuroSCORE index in assessing the risk of developing pressure injuries in cardiac surgery patients. The proposed hypotheses of the study are as follows:

**H1.** 
*Participants with a medium/high EuroSCORE II index have a higher risk of developing pressure injuries after surgery.*


**H2.** 
*The occurrence of pressure injuries after surgery affects the length of the patient’s overall recovery.*


## 2. Materials and Methods

### 2.1. Methods

A cross-sectional study was conducted on a sample of patients surgically treated at the Magdalena Cardiovascular Disease Clinic in the period from December 2022 to June 2023. The inclusion criteria were planned open heart surgery (coronary artery bypass graft (CABG), aortic valve replacement (AVR), mitral valve replacement (MVR) or mitral valve repair (PLMV), combined operations and other open heart surgery procedures), assessed EuroSCORE II index, and expressed consent to participate in the research. All patients admitted for other surgical and vascular surgical procedures, as well as those whose preoperative condition or the urgency of the procedure did not allow for timely information, were excluded from participation in the study. Our study has been approved by a suitably constituted Ethics Committee of the Clinic for Cardiovascular Diseases Magdalena under the number 195/INF-1502/23 and it conforms to the provisions of the Declaration of Helsinki in 1995 (as revised in Edinburgh 2000). Participants gave informed consent and patient anonymity was preserved.

### 2.2. Sample

To detect a medium effect size in the difference of continuous variables between two groups, with a significance level of 0.05 and power of 0.80, the required sample size was 128 participants (64 in each group). The present study included 211 participants in total, with 159 classified as low-risk and 59 as intermediate/high-risk according to EuroSCORE II, reflecting the actual distribution of risk categories in the study population.

Based on the EuroSCORE II index, the patients were divided into two groups—one with a low EuroSCORE II index (Group A) and a group with a medium/high EuroSCORE II index (Group B) according to the gradation published in the study by Silverborn et al. [41]. It is a validation study conducted with the aim of determining the performance of the EuroSCORE II index in patients undergoing cardiac bypass surgery. In this study, patients are classified into three groups according to the estimated risk: low < 4%, medium (intermediate) 4–8% and high > 8% risk. For the purposes of our study, patients were classified into two risk groups: low < 4% and medium/high > 4–8%. This distribution was chosen to achieve the largest possible sample of subjects in the group with a higher perioperative risk.

For all patients, an initial risk assessment for the occurrence of PI was performed upon admission (Braden 1) by ward nurses who had completed formal higher education and attended the hospital’s continuing education (CE) course on PI prevention. The same nurses also assessed individual characteristics (age, gender, body mass index (BMI)) and calculated the EuroSCORE II index before the operation. Data were collected for all patients on the type of procedure, use and duration of cardiopulmonary bypass (CPB) support, length of stay in the operating room and Intensive Care Unit (ICU), and total duration of hospitalisation. These data were collected using a specially created form. Monitoring and supervision of the skin for potential PI-related damage were carried out on the first, second and third postoperative days by recording the skin status in the exposed areas. For each patient, a postoperative assessment of the risk of PI (Braden 2) was performed on the first, second and third postoperative days by trained ICU nurses (CE course on PI prevention) or the ward nurses who performed the initial assessment. The Braden scale uses a gradation of scores based on the assessment of several basic risk factors for the development of PI, such as activity, mobility, nutrition, sensory perception, friction, shear and moisture [42]. Points range from 6 to 23, with a lower number indicating a higher risk of PI. Damage corresponding to the PI image, along with a detailed description of the changes, was reported by completing a standardised adverse event reporting form. The therapeutic measures chosen for the treatment of PI were also recorded on the same form.

### 2.3. Data Collection and Variable

Data were collected from patients’ medical records. The following categorical variables were included in the analysis: *Gender* (male/female); *EuroSCORE II risk category* (low risk < 4%, medium/high-risk ≥ 4%); *Therapy measures*, defined as interventions implemented to prevent the progression of PI, including patient repositioning, use of pressure-relieving surfaces, skin protection agents and application of specialized dressings; *Pressure injury (PI) stage*, classified according to the European Pressure Ulcer Advisory Panel (EPUAP 2019^5^) system: Stage 1 (non-blanchable erythema), Stage 2 (partial-thickness skin loss), Stage 3 (full-thickness skin loss), and Stage 4 (full-thickness tissue loss); *Surgical procedure type* (Coronary Artery Bypass Grafting(CABG), Aortic Valve Replacement(AVR), Mitral Valve Replacement or Mitral valve repair (MVR/PLMV), combination, other); *Surgery class* (elective, expedited, urgent). Continuous variables included age, body mass index (BMI), operative duration, cardiopulmonary bypass (CPB) total time, intensive care unit (ICU) total time, hospitalization duration, operating room total time, Braden score, and ejection fraction.

### 2.4. Data Analysis

Categorical variables are presented as absolute and relative frequencies. Differences between categorical variables were assessed using the Chi-square test, and effect sizes were quantified with Cramer’s V. The normality of distribution for continuous variables was evaluated with the Shapiro–Wilk test. Since the data did not follow a normal distribution, continuous variables are presented as median and interquartile range (IQR). Group differences in continuous variables according to EuroSCORE II risk category were analysed using the Mann–Whitney U test, with standardized *Z* values, Hodges–Lehmann estimates of median difference, and corresponding 95% confidence intervals reported. To examine predictors of pressure injury occurrence, bivariate and multivariate logistic regression analyses were performed. Variables with *p* < 0.10 in bivariate analysis were entered into the multivariate model. The multivariate model was adjusted for age, BMI, and operative duration. Multicollinearity among predictors was assessed using the variance inflation factor (VIF), and no significant multicollinearity was detected (all VIF < 5). Odds ratios (OR) with 95% confidence intervals (CI) were calculated to quantify associations. All *p*-values are two-tailed, with a significance level set at *α* = 0.05. Statistical analyses were performed using MedCalc^®^ Statistical Software version 23.3.7 (MedCalc Software Ltd., Ostend, Belgium; Available online: https://www.medcalc.org; (accessed on 29 October 2025).

## 3. Results

The individual characteristics of the participants according to the observed parameters are shown in Table 1. The sample consisted of predominantly male participants (n = 149; 70.6%), with only one third being females (n = 62; 29.4%). The majority of participants were elderly patients, with a median age of 69 years (IQR: 59–75) but with some variation in the age of the whole sample (R: 33–85). The median BMI of the participants was 28.73 kg/m^2^, classifying the sample as moderately overweight (IQR: 25.78–32.41).

Table 2 presents the results of the comparison between the EuroSCORE II index and the observed characteristics of the participants, as well as the estimated risk for the occurrence of PI. The results show that the median age of participants in Group B was 73 years (IQR: 68–77), while in Group A it was 66 years (IQR: 58–73), indicating that participants with a medium or high EuroSCORE II index were statistically significantly older (*p* < 0.001).

When assessing the risk of PI before surgery (Braden 1), the calculated values in Group A showed a median score of 20 points (IQR: 19–21), similar to Group B, whose median preoperative score was also 20 points but with a wider range of results (IQR: 17–21). The difference in estimated preoperative risk between the two groups was statistically significant (*p* = 0.011), indicating that patients with medium or higher EuroSCORE II risk had a higher risk of PI according to the Braden scale.

The cardiac ejection fraction (EF) in the total sample had a median value of 60% (IQR: 50–65), indicating values within the normal range. However, a wide dispersion of estimated values (R: 15–75%) suggests that the sample included participants with significantly impaired heart function, while the majority had higher, normal function.

The time spent with CPB support in Group A averaged 1.3 hours (IQR: 1.11–1.57), whereas in Group B the average time was significantly greater at 1.92 hours (IQR: 1.41–2.61), indicating a more complex surgical course. Assessment of the total time spent in the operating room showed similar trends. Participants in Group A spent an average of 5 h in the operating room (IQR: 4–5), while those in Group B spent 6 hours (IQR: 5–7), demonstrating that surgical procedures in participants with a medium/high EuroSCORE II index lasted longer. The time spent in the ICU was also statistically significantly different (*p* < 0.001), with participants with a medium/high EuroSCORE II index spending an average of 69 hours (IQR: 47–119) in the ICU. The overall duration of hospitalisation showed a prolonged stay for participants in Group B, who stayed an average of 14 days (IQR: 10–21), compared with participants in Group A, whose hospital stay was 3 days shorter on average (M: 10; IQR: 9–12).

Table 3 presents the results of a comparison of individual characteristics, the estimated EuroSCORE II index, and the incidence of PI during the observed period. Considering the type of procedure, a statistically significant difference (*p* < 0.001) was observed between the two groups of participants, with patients with a medium or high EuroSCORE II index more frequently hospitalised for complex open-heart procedures.

The assessment of skin discolouration indicating the presence of PI showed that most participants in both groups had no signs of skin damage on the first postoperative day. Further evaluation revealed that on the second postoperative day, a number of participants (n = 7; 13.5%) in Group B developed skin damage in the form of scars, bruises, or haematomas, resulting in a statistically significant difference in assessed skin status (*p* = 0.002) compared to Group A, but without a direct clinical relation to the emergence of PI.

The final evaluation of skin status showed that changes corresponding to pressure injury after the operative procedure were observed in 30.8% (n = 16) of participants in Group B, with a statistically significant difference (*p* = 0.043). According to the defined degree of observed skin damage, Stage 2 was reported in 9.6% of participants in Group B. The association between the presence and stage of pressure injury is statistically significant, with a modest practical impact (Cramér’s V = 0.14–0.19). Analysis of the localisation of these changes showed they were most often present in the gluteal region and on the heels.

In the bivariate logistic regression analysis, age (*p* = 0.002), EuroSCORE II (*p* = 0.001), EuroSCORE II medium/high-risk category (*p* = 0.045), ejection fraction (*p* < 0.001), and combined surgical procedures (*p* = 0.022) were significantly associated with the occurrence of pressure injury. However, after adjustment for age, BMI, and operative duration in the multivariate model, only ejection fraction remained an independent predictor (OR = 0.97, 95% CI 0.94–0.98, *p* = 0.036) (Table 4).

## 4. Discussion

The primary aim of this study was to determine the association between EuroSCORE II index values and the occurrence of PI in cardiac surgery patients. The occurrence of PI after cardiac surgery is a serious complication that can significantly alter the course of a patient’s treatment. The study showed that the presence of PI in patients with elevated perioperative EuroSCORE II index values was almost half as high as in patients with a lower risk score. More importantly, the results indicated that patients with a higher EuroSCORE index were more likely to develop more severe tissue damage in the form of stage II PI.

The occurrence of PI in patients with an elevated EuroSCORE II risk remains the subject of a limited number of studies. A study conducted by Ettema et al. [43], aimed at investigating the preoperative profile of patients with an increased risk for postoperative delirium, depression, pressure ulcers and infections, showed that EuroSCORE II index values can be associated with postoperative PI, but with the addition of other risk parameters such as tricuspid insufficiency, serum creatinine levels and the use of low molecular weight heparin. A more recent study conducted by Ayazoglu et al. [44], with the aim of determining risk factors for the occurrence of PI after the procedure, determined that an elevated EuroSCORE II index (>4), along with other risk factors (advanced age, comorbidities, length of surgery), can be considered a proven risk factor for the occurrence of PI in the early postoperative period. Our study confirmed these findings with the addition of the fact that an elevated EuroSCORE II index certainly assumes risk for the development of serious skin damage in deeper skin layers on pronounced points of the body in the early recovery period. Moreover, ejection fraction (EF) emerged as the only independent predictor of pressure injuries, indicating that patients with poorer cardiac function are at higher risk for developing PI following cardiac surgery. Although the association between reduced ejection fraction and the occurrence of respiratory, neurological, and renal complications, as well as overall mortality in cardiac patients, has been previously described [45,46,47], studies to date have not investigated its relationship with the development of pressure injuries following cardiac surgery.

It is important to emphasise that the results showed an association between elevated EuroSCORE II values and the development of injuries in deeper tissue layers, indicating the occurrence of stage 2 pressure injuries. Although a study by Alderden et al. [48]. found different mechanisms of formation, risk factors, and presentation sites for deep tissue pressure injuries (DTPI) and stage 2 PI in intensive care patients, our study has shown that more severe tissue injuries in cardiac surgery patients occur in areas particularly exposed to prolonged pressure, such as the heels and gluteal region. Elevated EuroSCORE II values in patients with deeper tissue damage indicate a range of intrinsic risk factors that reduce tissue perfusion. Combined with environmental and situational factors, these contribute to the development of pressure injuries. Therefore, it is important to consider parameters such as the EuroSCORE II index, which can significantly assist in identifying intrinsic risk factors with a low degree of modifiability.

The EuroSCORE II assessment model was primarily developed to determine the risk of perioperative mortality in cardiac surgery. Subsequent research has shown that this parameter can also be used to monitor other treatment outcome indicators, such as specific postoperative complications, length of ICU stay, duration of mechanical ventilation, and total length of hospitalisation [49,50,51,52,53,54]. A study by Hirose et al. [51]. confirmed that the EuroSCORE II index is well correlated with total mortality after the procedure, the incidence of major postoperative complications (heart failure, renal dysfunction, stroke, pneumonia, mediastinitis), and three recovery time parameters: intubation time, ICU length of stay, and total postoperative hospital stay. A study by Goeber et al. [55]. also confirmed that patients with an elevated EuroSCORE II index require more care and adjustment of postoperative rehabilitation. Our study found identical results, clearly showing that patients in the high-risk group had longer operative times, CPB support, ICU stays, and total hospitalisation duration. Many studies have confirmed the association between the occurrence of pressure injuries (PIs) and prolonged ICU stay and overall hospitalisation, with a significant impact on treatment costs [56,57,58]. Despite the fact that these variables are considered independent risk factors for the development of pressure injuries (PI), it is very difficult to clarify the cause-and-effect relationship. Therefore, it is important to emphasise that elevated EuroSCORE II values clearly indicate a potential risk of prolonged hospitalisation, including longer surgical duration and length of stay in the ICU, which may independently represent risk factors for the development of pressure injuries.

Identifying patients at increased risk of PI before surgery remains challenging, primarily due to various specific factors that contribute to its occurrence, such as the use of vasoactive drugs, mechanical ventilation, risk of bleeding, duration of the surgical procedure, and prolonged inability to change body position. The value of the EuroSCORE II index lies in its assessment of several intrinsic risk parameters for mortality during and after surgery, which can undoubtedly influence the occurrence of various postoperative complications, including PI. Although we did not analyse each of these parameters individually in our study, the results clearly show that an elevated EuroSCORE II index affects the appearance of skin changes that, as early as the second postoperative day, result in the development of PI, usually Stage 1.

Recognizing and understanding potential risk factors is key to defining targeted prevention strategies [59]. PI risk assessment tools, such as the Braden scale, which have been generally accepted and widely used, have shown weak to moderate predictive value with limited specificity and sensitivity [60,61,62]. The Braden scale was also of limited value in our case, as all patients in our study were assessed as being at risk, without a clear gradation of risk within the group. The results of our research clearly indicate the need to re-evaluate such tools, especially for use in specific patient groups. Therefore, we believe that using the EuroSCORE II index for this purpose in patients undergoing cardiac surgery can be clinically justified. It is important to note that this risk assessment tool clearly reflects the complexity of the patient’s condition, which can improve timely predictions of the need for treatment, care, and length of hospitalisation. Timely application can significantly improve nursing practice focused on preventive measures, such as implementing aids for the prevention of pressure injuries during and immediately after surgery, mobilisation, and monitoring of skin condition in the early postoperative period.

The present study has certain limitations. It was conducted on a limited sample of patients, with the number determined by the total available surgical patients who met the inclusion criteria during a specific period. Additionally, the influence of further risk factors, such as anaemia, incontinence, or other chronic health conditions, was not considered. The study also did not assess the impact of other intraoperative and intrinsic factors, such as haemodynamic stability, hypothermia, nutrition, albumin level, and the use of vasoactive drugs. Furthermore, the same standard preventive measures were applied to all patients in the sample, regardless of the estimated preoperative risk for the occurrence of PI, and it is not currently possible to determine their influence on the final results. Finally, the study design did not allow for the determination of causal relationships between the observed variables. Therefore, we recommend further studies focusing on the critical factors identified here and their influence on the occurrence of PI in cardiac surgery patients.

Potential sources of bias. As this was a cross-sectional study, selection and information bias could not be completely avoided. Data were extracted from medical records, which may contain inconsistencies in documentation. Furthermore, certain unmeasured variables, such as intraoperative management techniques, nursing workload, or postoperative skin care practices, may have acted as residual confounders. Nevertheless, the inclusion of age, BMI, and operative duration as covariates in the multivariate model helped to reduce the impact of potential confounding.

## 5. Conclusions

The occurrence of PI in patients undergoing cardiac surgery remains a complex issue without a definitive answer. The specifics of the surgical procedure and the complexity of the patient’s condition often combine to increase the risk of various complications that may threaten recovery and treatment outcomes. Timely identification of patients at increased risk of developing PI in the early postoperative period remains the preferred approach. The use of EuroSCORE II for early detection of patients at higher risk of developing PI has proved useful and informative. We recommend further research on this topic, including additional specific risk parameters and examining their influence and interrelation with the occurrence of PI. We believe that improving PI preventive strategies can significantly contribute to the overall management of adverse events and complications following complex surgical procedures, with positive implications for the treatment course and length of hospitalisation.

## Figures and Tables

**Table 1 healthcare-13-02880-t001:** Descriptive statistics.

	n	Median (IQR)	Range
Gender [n (%)]			
M	149 (70.6)		
F	62 (29.4)		
Age [years]	211	69 (59–75)	33–85
BMI [kg/m^2^]	211	28.73 (25.78–32.41)	19.27–49.95
EuroSCORE II	211	1.86 (1.17–3.91)	0.55–55.59
BRADEN 1	211	20 (19–21)	0–22
BRADEN 2	211	15 (13–16)	0–19
Ejection fraction [%]	211	60 (50–65)	15–75
CPB total time [h]	211	1.37 (1.12–1.80)	0–4.40
Op. room total time [h]	211	5 (4–6)	3–10
ICU total time [h]	211	47 (41–70)	1.49–479
Hospitalization [days]	211	10 (9–13)	7–66

Source: Medical documentation 2022–2023. Abbreviations: M—Males; F—Females; BMI—Body Mass Index; CPB—Cardiopulmonary Bypass; Op.—Operating; ICU—Intensive Care Unit.

**Table 2 healthcare-13-02880-t002:** Relationship between continuous variables and the EuroSCORE II risk index (Group A = low EuroSCORE II index; Group B = medium/high EuroSCORE II index).

EuroSCORE II	Group	n	Median (IQR)	^†^ Diffe Rence	95% CI	*z* Statistic	*p* * Value
Overall	Males	149	1.78 (1.09–4.02)	0.30	−0.10 to 0.74	−1.57	0.117
	Females	62	2.39 (1.38–3.80)				
Males	A	111	1.33 (0.96–2.04)	5.60	4.83 to 6.90	−9.18	<0.001
	B	38	7.11 (5.53–10.99)				
Females	A	48	1.64 (1.30–2.89)	6.39	4.51 to 11.09	−5.66	<0.001
	B	14	8.41 (5.68–14.70)				
EuroSCORE II	A	159	1.48 (1.04–2.43)	5.72	4.98 to 6.94	−10.82	<0.001
	B	52	7.54 (5.68–12.26)				
Age [years]	A	159	66 (58–73)	6	3 to 9	−3.73	<0.001
	B	52	73 (68–77)				
BMI [kg/m^2^]	A	159	29.05 (25.81–32.32)	−0.94	−2.45 to 0.86	−1.01	0.315
	B	52	26.99 (25.78–32.96)				
BRADEN 1	A	159	20 (19–21)	−1	−1 to 0	−2.56	0.011
	B	52	20 (17–21)				
BRADEN 2	A	159	15 (13–16)	−1	−2 to 0	−2.59	0.009
	B	52	14 (13–15)				
Ejection fraction [%]	A	159	65 (55–65)	−5	−10 to 0	−3.49	<0.001
	B	52	55 (38–65)				
CPB total time [h]	A	159	1.3 (1.11–1.57)	0.53	0.30 to 0.75	−4.78	<0.001
	B	52	1.92 (1.41–2.61)				
Op. room total time [h]	A	159	5 (4–5)	1	1 to 2	−5.31	<0.001
	B	52	6 (5–7)				
ICU total time [h]	A	159	46 (27–66)	24	20 to 43	−5.49	<0.001
	B	52	69 (47–119)				
Hospitalization [days]	A	159	10 (9–12)	3	2 to 5	−4.99	<0.001
	B	52	14 (10–21)				

Source: Medical documentation 2022–2023. * Mann–Whitney U test; ^†^ Hodges–Lehmann median difference. Abbreviations: IQR—interquartile range; 95% CI—95% Confidence Interval; CPB—Cardiopulmonary Bypass; ICU—Intensive Care Unit. Bold font denotes statistically significant differences. Each Z statistic corresponds to an independent Mann–Whitney U test comparing Group A and Group B within the specified subgroup. Z—standardized test statistic reported by MedCalc instead of the raw U value.

**Table 3 healthcare-13-02880-t003:** Patient characteristics and occurrence of pressure injury according to the EuroSCORE II index.

		Number (%) Participants			*p* *	Effect Size (Cramer’s V)
		Low EuroSCORE II Index (*n* = 159)	Medium/High EuroSCORE II Index (*n* = 52)	Total (*n* = 211)		
Gender	Male	111 (69.8)	38 (73.1)	149 (70.6)	0.654	0.031
	Female	48 (30.2)	14 (26.9)	62 (29.4)		
Surgical procedures	CABG	48 (30.4)	11 (21.2)	59 (28.1)	<0.001	0.427
	AVR	57 (36.1)	12 (23.1)	69 (32.9)		
	MVR + PLMV	29 (18.4)	0	29 (13.8)		
	Combination	22 (13.9)	27 (51.9)	49 (23.3)		
	Other	2 (1.2)	2 (3.8)	4 (1.9)		
Surgery class	Elective	125 (78.6)	19 (36.5)	144 (68.2)	<0.001	0.395
	Expedited	31 (19.5)	28 (53.9)	59 (28)		
	Urgent	3 (1.9)	5 (9.6)	8 (3.8)		
CPB support	Yes	156 (98.1)	51 (98.1)	207 (98.1)	0.978	0.001
	No	3 (1.9)	1 (1.9)	4 (1.9)		
Postoperative Day 1	Clean	136 (85.5)	40 (76.9)	176 (83.4)	0.209	0.147
	Redness	14 (8.8)	10 (19.2)	24 (11.4)		
	Blister	1 (0.6)	0 (0)	1 (0.5)		
	Other	8 (5)	2 (3.8)	10 (4.7)		
Postoperative Day 2	Clean	132 (83)	33 (63.5)	165 (78.2)	0.002	0.245
	Redness	23 (14.5)	12 (23.1)	35 (16.6)		
	Other	4 (2.5)	7 (13.5)	11 (5.2)		
Postoperative Day 3	Clean	132 (83)	35 (67.3)	167 (79.1)	0.051	0.168
	Redness	23 (14.5)	15 (28.8)	38 (18)		
	Other	4 (2.5)	2 (3.8)	6 (2.8)		
Pressure Injury	Yes	28 (17.6)	16 (30.8)	44 (20.9)	0.043	0.140
	No	131 (82.4)	36 (69.2)	167 (79.1)		
PI Stage	Without	131 (82.4)	36 (69.2)	167 (79.1)	0.021	0.191
	^†^ Stage 1	25 (15.7)	11 (21.2)	36 (17.1)		
	^†^ Stage 2	3 (1.9)	5 (9.6)	8 (3.8)		
Localization	Without	131 (82.4)	36 (69.2)	167 (79.1)	0.058	0.164
	Gluteal region	27 (17)	14 (26.9)	41 (19.4)		
	Heels	1 (0.6)	2 (3.8)	3 (1.4)		
Therapy measures	Without	0	37 (71.2)	37 (17.5)	<0.001	0.820
	Heeling cream	27 (17)	8 (15.4)	35 (16.6)		
	Hydrocolloid dressing	132 (83)	7 (13.5)	139 (65.9)		

Source: Medical documentation 2022–2023. Abbreviations: CPBCardiopulmonary Bypass; Bold font denotes statistically significant differences. CABG—Coronary Artery Bypass Graft; AVR—Aortic Valve Replacement; MVR + PLMV—Mitral Valve Replacement + Mitral Valve Repair. * Chi-square test. ^†^ Stage 1 = non-blanchable erythema; Stage 2 = partial-thickness skin loss.

**Table 4 healthcare-13-02880-t004:** Prediction of the probability of developing pressure injury (bivariate and multivariate logistic regression).

	*ß*	Wald	*p* Value	OR	95% CI
Bivariate logistic regression					
Gender (F)	0.15	0.16	0.690	1.16	0.56 to 2.37
Age	0.06	9.85	0.002	1.06	1.02 to 1.10
BMI	−0.03	0.65	0.422	0.97	0.91 to 1.04
Operative duration	0.17	1.67	0.201	1.18	0.92 to 1.52
EuroSCOR II	0.10	10.51	0.001	1.11	1.04 to 1.18
EuroSCORE II (medium/high)	0.73	4.01	0.045	2.08	1.02 to 4.26
Ejection fraction	−0.04	11.20	<0.001	0.96	0.93 to 0.98
CPB total time	−18.8	0	>0.99	-	-
Surgical procedures (CABG)					
AVR	−0.30	0.40	0.528	0.74	0.29 to 1.89
MVR + PLMV	−0.69	0.97	0.323	0.50	0.13 to 1.96
Combination	1.02	5.23	0.022	2.76	1.16 to 6.61
Other	0.37	0.09	0.755	1.45	0.14 to 15.35
Surgery class (elective)					
Expedited	0.69	3.75	0.053	1.99	0.99 to 4.01
Urgent	−19.6	0	0.998	-	-
Multivariate logistic regression					
Ejection fraction	−0.03	3.99	0.036	0.97	0.94 to 0.98
Constant	−2.94	1.96	0.16		

*β*—regression coefficient; OR—odds ratio; CI—confidence interval. CABG—Coronary Artery Bypass Graft; AVR—Aortic Valve Replacement; MVR + PLMV—Mitral Valve Replacement + Mitral Valve Repair. Bold font denotes statistically significant differences. Multivariate model adjusted for age, BMI, and operative duration. Reference categories: male gender, low EuroSCORE II risk, CABG procedure, elective surgery class.

## Data Availability

The original contributions presented in this study are included in the article. Further inquiries can be directed to the corresponding author(s).

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
