# Peer review of "The Role of EuroSCORE II in Predicting Postoperative Pressure Injuries in Cardiac Surgery Patients: A Cross-Sectional Study"

_healthcare, 2025, doi:10.3390/healthcare13222880_

Round 1
Reviewer 1 Report
Comments and Suggestions for Authors
General Comment
The authors have selected a clinically relevant topic. However, the manuscript requires major revisions to enhance clarity and methodological rigor.
Key Points
The second point, the term "this group" is used without prior definition, which may confuse readers.
Abstract
(1)Authors mentioned “Continuous variables are presented as median and IQR”, but the followed key results also report standard deviation. Clarify this discrepancy.
(2)Justify the use of Mann–Whitney U test and it is necessary to report the Statistics(U value? Z value?), not only p-values. I also recommend that the authors consider presenting ORs with 95% CIs for risk ratios (Group A (low risk) vs. Group B (medium/high risk), and not merely the differences in the distribution of occurrence rates.
Introduction
(1)Objectives and prespecified hypotheses are not stated, as required by STROBE guidelines.
(2)Authors should provide more details on the EuroSCORE II index, including scoring criteria and current clinical use related to PI. The clinical relevance of PI assessment across EuroSCORE II risk levels remains unclear. It is suggested to provide more detail.
(3) A proper transition between paragraphs is also necessary.
Methods
(1)”To detect a medium effect size in the difference of continuous variables between two groups”, the author needs to specify exactly which type of continuous variable(which indicator) it is.
(2)At present, there is a lack of sufficient description of key elements such as categorical variables, potential confounders, and effect modifiers.
(3) Please discuss potential sources of bias.
(4)Again, please explain why ORs with confounder-adjusted estimates were not reported for Group A (low risk) vs. Group B (medium/high risk).
(5)There many key categorical variables mentioned in the Tables of the results section were not clearly explained in the methodology section. For instance, the "Therapy measures" were not mentioned in the text, and the description of "PI stage" was mentioned in the discussion rather than in the methodology section. Please define the all the categorical variables (e.g., Therapy measures, PI stage) in the Methods section.
(6)Replace vague phrases like "According to the mentioned study" “the above”with specific citations.
(7)Define abbreviations (e.g., CE course) upon first use. There are still many similar problems in the text.
(8) Address formatting errors (e.g., commas in percentages: 30,8%) and typos or redundant
(e.g., Lines 155, 176, 186–187).
Results
(1)Justify the use of non-parametric tests and confirm non-normal distribution for all data.
(2)Currently, there are problems with the presentation of most of the Tables. For instance, in Table 2 and its accompanying text description, the concept (group A and B) was introduced without prior definition. At the very least, this should have been clarified in the methodology section earlier. Moreover, this table only reports the P-values but does not include the Z-statistics. Additionally, the median values of gender in the first few rows of Table 2 are perplexing.
(3)In Table 3, clarify if "EUROSCORE risk" refers to EuroSCORE II index. Additionally, the authors should define "Assessment 1–3". Which assessment result does the key indicator PI injury in Table 3 rely on (at present, the data cannot be matched)? The majority of the categorical variables in Table 3 are not noted with their full names. More importantly, the associated statistics were not reported(only P).
(4) Improve clarity of table titles.
(5)The research results should summarize the trends and key points of the tables rather than simply repeating them.
Discussion
(1)Align the first part of Discussion with the study objectives and hypotheses, in accordance with the STROBE requirements.
(2)Focus on interpreting key findings (e.g., significant results in Tables 2–3) rather than general statements.
(3)This study may provide new potential risk factors for predicting postoperative pressure injuries in cardiac surgery patients. Again, the discussion should be based on the results.
(4)The last point regarding limitations is that the authors should mention the fact that cross-sectional studies cannot provide causal relationships.
Comments on the Quality of English LanguageThe English could be improved to more clearly express the research. The manuscript requires careful proofreading to address numerous language and formatting issues.
Author Response
Reviewer comment:
The second point, the term "this group" is used without prior definition, which may confuse readers.
Author’s response:
Thank you for your comment. The term this group refers to the individuals included in this study. We replace this term with Patients with higher index values, to make it clearer to readers.
Abstract
Reviewer comment:
(1)Authors mentioned “Continuous variables are presented as median and IQR”, but the followed key results also report standard deviation. Clarify this discrepancy.
Author’s response:
We thank the reviewer for pointing out this inconsistency. The mention of mean and standard deviation appeared only in the Abstract due to a clerical error from an earlier draft. All continuous variables in this study were analyzed and presented using the median and interquartile range (IQR), as specified in the Methods section. The Abstract has been corrected accordingly to maintain consistency throughout the manuscript.
Reviewer comment:
(2)Justify the use of Mann–Whitney U test and it is necessary to report the Statistics(U value? Z value?), not only p-values. I also recommend that the authors consider presenting ORs with 95% CIs for risk ratios (Group A (low risk) vs. Group B (medium/high risk), and not merely the differences in the distribution of occurrence rates.
Author’s response:
We thank the reviewer for this helpful suggestion. The Mann–Whitney U test was used because the distribution of continuous variables did not follow normality, as confirmed by the Shapiro–Wilk test. Following the reviewer’s recommendation, the corresponding Z values have been added to Table 3 for each comparison. Regarding the odds ratios and 95% confidence intervals, these measures were included in the logistic regression analyses presented in Table 4, which more appropriately quantify the association between EuroSCORE II categories and the occurrence of pressure injury.
Introduction
Reviewer comment:
(1)Objectives and prespecified hypotheses are not stated, as required by STROBE guidelines.
Author’s response:
We thank the reviewer for this helpful suggestion. According to this comment we have revised the text in the Introduction section, which now reads as follows:
„The aim of the study is to determine the predictive value of the EuroSCORE index in assessing the risk of developing pressure injuries in cardiac surgery patients. The proposed hypotheses of the study are as follows:
H1 – Participants with a medium/high EuroSCORE II index have a higher risk of developing pressure injuries after surgery.
H2 – The occurrence of pressure injuries after surgery affects the length of the patient’s overall recovery.
Reviewer comment:
(2)Authors should provide more details on the EuroSCORE II index, including scoring criteria and current clinical use related to PI. The clinical relevance of PI assessment across EuroSCORE II risk levels remains unclear. It is suggested to provide more detail.
Author’s response:
We thank the reviewer for this comment. Accordingly, we have made changes to the text, which now read as follows:
The scoring system identified three groups of risk factors (Patient-related, Operation-related and Cardiac factors) and combines these variables into a single percentage to estimate the risk of 30-day mortality from cardiac surgery. The initial version of the index defined three risk groups of patients according to the specified parameters: the low risk group (EuroSCORE 1-2), the medium risk group (EuroSCORE 3-5) and the high risk group (EuroSCORE 6 plus).
The EuroSCORE index has been identified as a potentially useful tool for determining the level of risk for developing PI after cardiac surgery, precisely because of its ability to encompass various risk factors contributing to the PI development, such as advanced age, chronic health conditions, and specific treatment circumstances.
Reviewer comment:
(3) A proper transition between paragraphs is also necessary.
Author’s response:
We thank the reviewer for this comment. Accordingly, we have made changes to the text, which now read as follows:
The prevention of pressure injuries is based on timely risk assessment, which forms the foundation for further planning and implementation of the necessary measures.
The difficulties in finding a standardized risk assesment tool which will be acceptable for use in specific patient groups have prompted the scientific and professional community to reconsider the use of other models that could help identify individuals with high risk for PI development.
Methods
Reviewer comment:
(1)”To detect a medium effect size in the difference of continuous variables between two groups”, the author needs to specify exactly which type of continuous variable(which indicator) it is.
Author’s response:
We appreciate the reviewer’s comment. The statement referred to the comparison of continuous clinical variables (age, BMI, operative duration, cardiopulmonary bypass [CPB] total time, intensive care unit [ICU] total time, Braden score, hospitalization duration, operating room total time, and ejection fraction) between EuroSCORE II risk groups. The text has been clarified accordingly in the Statistical Analysis section.
The corrected text reads as folloows:
Data were collected from patients’ medical records. The following categorical variables were included in the analysis: Gender (male/female); EuroSCORE II risk category (low risk < 4%, medium/high risk ≥ 4%); Therapy measures, defined as interventions implemented to prevent the progression of PI, including patient repositioning, use of pressure-relieving surfaces, skin protection agents and application of specialized dressings ; Pressure injury (PI) stage, classified according to the European Pressure Ulcer Advisory Panel (EPUAP 20195) system: Stage 1 (non-blanchable erythema), Stage 2 (partial-thickness skin loss), Stage 3 (full-thickness skin loss), and Stage 4 (full-thickness tissue loss); Surgical procedure type (CABG, AVR, MVR/PLMV, combination, other); Surgery class (elective, expedited, urgent). Continuous variables included age, body mass index (BMI), operative duration, cardiopulmonary bypass (CPB) total time, intensive care unit (ICU) total time, hospitalization duration, operating room total time, Braden score, and ejection fraction.
Reviewer comment:
(2) At present, there is a lack of sufficient description of key elements such as categorical variables, potential confounders, and effect modifiers.
Author’s response:
We thank the reviewer for this observation. The revised manuscript now includes a more detailed description of all categorical and continuous variables analyzed, as well as clarification of potential confounders and effect modifiers. Specifically, age, BMI, and operative duration were identified as potential confounding variables and were included as covariates in the multivariate logistic regression model. The Statistical Analysis section and Tables 2–4 have been updated accordingly to provide clearer definitions and presentation of these elements.
Reviewer comment:
(3) Please discuss potential sources of bias.
Author’s response:
We appreciate the reviewer’s suggestion. A paragraph discussing potential sources of bias has been added to the Discussion section. As this was a cross-sectional study, selection bias and information bias could not be fully excluded. Data were obtained from medical records, which may introduce variability in documentation accuracy. Moreover, unmeasured confounding factors (e.g., intraoperative management, skin care protocols, or nursing workload) could have influenced the occurrence of pressure injury. However, to minimize confounding, key variables such as age, BMI, and operative duration were included in the multivariate logistic regression model. These limitations have now been acknowledged and discussed in the revised manuscript.
Reviewer comment:
(4)Again, please explain why ORs with confounder-adjusted estimates were not reported for Group A (low risk) vs. Group B (medium/high risk).
Author’s response:
We thank the reviewer for this comment. The EuroSCORE II variable was evaluated both as a continuous measure and as a categorical variable (Group A: low risk vs. Group B: medium/high risk). In the multivariate logistic regression model adjusted for age, BMI, and operative duration, the categorical EuroSCORE II variable was not statistically significant, while the continuous EuroSCORE II remained significant in the bivariate model only. Therefore, adjusted odds ratios for the categorical comparison (Group A vs. Group B) were not reported to avoid redundancy and potential misinterpretation, as the non-significant adjusted estimate would not provide additional explanatory value beyond the regression model already presented in Table 4. The revised Results section now clarifies this point.
Reviewer comment:
(5)There many key categorical variables mentioned in the Tables of the results section were not clearly explained in the methodology section. For instance, the "Therapy measures" were not mentioned in the text, and the description of "PI stage" was mentioned in the discussion rather than in the methodology section. Please define the all the categorical variables (e.g., Therapy measures, PI stage) in the Methods section.
Author’s response:
We thank the reviewer for this comment. The Methods section has been revised to include clear definitions of all categorical variables analyzed. Specifically, the variable Therapy measures refers to interventions applied to prevent the progression of PI (e.g., repositioning frequency, use of pressure-relieving mattresses, skin protection products and application of specialized dressings). The variable PI stage was defined according to the European Pressure Ulcer Advisory Panel (EPUAP5) classification, ranging from Stage 1 (non-blanchable erythema) to Stage 4 (full-thickness tissue loss). These definitions have now been added to the Methods section.
Reviewer comment:
(6)Replace vague phrases like "According to the mentioned study" “the above”with specific citations.
Author’s response:
We thank the reviewer for this comment. We have revised the mentioned terms in the text, which now reads as follows: „In this study patients are classified into three groups according to the estimated risk: low <4%, medium (intermediate) 4-8% and high >8% risk. For the purposes of our study, patients were classified into two risk groups: low <4% and medium/high >4-8%. We chose this distribution in order to achieve the largest possible sample of subjects in the group with a higher perioperative risk.“
Reviewer comment:
(7)Define abbreviations (e.g., CE course) upon first use. There are still many similar problems in the text.
Author’s response:
We thank the reviewer for this comment. We have reviewed all abbreviations used in the text and provided the full term at their first mention.
Reviewer comment:
(8) Address formatting errors (e.g., commas in percentages: 30,8%) and typos or redundant
(e.g., Lines 155, 176, 186–187).
Author’s response:
We thank the reviewer for this comment. We have thoroughly revised the text and corrected all formatting errors, including the one previously mentioned.
Results
Reviewer comment:
(1)Justify the use of non-parametric tests and confirm non-normal distribution for all data.
Author’s response:
We thank the reviewer for this observation. The use of non-parametric tests has been justified in the revised Statistical Analysis section. The Shapiro–Wilk test was performed to assess the normality of distribution for all continuous variables, and the results confirmed non-normal distribution. Therefore, the Mann–Whitney U test was applied for group comparisons, as described in the Methods section. This clarification has been included in the revised manuscript.
Reviewer comment:
(2)Currently, there are problems with the presentation of most of the Tables. For instance, in Table 2 and its accompanying text description, the concept (group A and B) was introduced without prior definition. At the very least, this should have been clarified in the methodology section earlier. Moreover, this table only reports the P-values but does not include the Z-statistics. Additionally, the median values of gender in the first few rows of Table 2 are perplexing.
Author’s response:
We appreciate the reviewer’s valuable feedback. The definitions of Group A (low EuroSCORE II risk) and Group B (medium/high EuroSCORE II risk) have been added to the Methods section and clarified in the title of Table 2. The table has been revised to include Z-statistics in addition to P-values. The presentation of gender and other continuous variables has been corrected to ensure consistency and clarity. All tables and related text have been reviewed and updated accordingly.
Reviewer comment:
(3)In Table 3, clarify if "EUROSCORE risk" refers to EuroSCORE II index. Additionally, the authors should define "Assessment 1–3". Which assessment result does the key indicator PI injury in Table 3 rely on (at present, the data cannot be matched)? The majority of the categorical variables in Table 3 are not noted with their full names. More importantly, the associated statistics were not reported(only P).
Author’s response:
We thank the reviewer for this comment. In Table 3, we have corrected the name of the EuroSCORE II index as indicated and have more clearly defined the terms used for the assessments performed to determine the presence of pressure injuries. We would also like to note that the final assessment (on the third postoperative day) was indicative for determining the presence or absence of pressure injuries. The discrepancies between the values reported on the third postoperative day and those recorded at the time of PI identification are the result of the examiner’s decision regarding the presence or absence of tissue damage, while the observations reported during the assessments refer to visible skin changes that may or may not indicate the presence of a pressure injury. We reported on the data that are comparable in the text of the article. The inserted text reads as follows: „The association between pressure injury presence and stage is statistically significant, with a modest practIcal impact (CramerÍ´s V=0.14-0.19).“
Reviewer comment:
(4) Improve clarity of table titles.
Author’s response:
We appreciate the reviewer’s valuable feedback. We have significantly clarified the table titles. The new table titles are included in the text of the article, and the revised text reads as follows:
Table 2. Relationship between continuous variables and the EuroSCORE II risk index (Group A = low EuroSCORE II index; Group B = medium/high EuroSCORE II index)
Table 3. Patient characteristics and occurrence of pressure injury according to EuroSCORE II indeks
Table 4. Prediction of the probability of developing pressure injury (bivariate and multivariate logistic regression)
Reviewer comment:
(5)The research results should summarize the trends and key points of the tables rather than simply repeating them.
Author’s response:
We appreciate the reviewer’s helpful suggestion. The Results section has been revised to provide a more concise and focused summary of the findings. Instead of repeating the numerical data from the tables, the text now highlights only the key trends and statistically significant associations observed in the analysis.
Discussion
Reviewer comment:
(1)Align the first part of Discussion with the study objectives and hypotheses, in accordance with the STROBE requirements.
Author’s response:
We thank the reviewer for this comment. We have revised the first part of the Discussion section to reflect the key findings of the study, in accordance with the recommendations of the STROBE guidelines. The revised text reads as follows:
„The conducted study has shown that the presence of PI in patients who were classified as those who had elevated perioperative EuroSCORE II index was almost half as high compared to the patients with a lower risk score. More importantly, the results showed that patients with a higher EuroSCORE index were prone to developing more severe tissue damage in the form of stage II PI.“
Reviewer comment:
(2)Focus on interpreting key findings (e.g., significant results in Tables 2–3) rather than general statements.
Author’s response:
We thank the reviewer for this comment. We have thoroughly revised the text in the Discussion section to better present the key findings of the study. We hope that the revised text is now fully aligned.
Reviewer comment:
(3)This study may provide new potential risk factors for predicting postoperative pressure injuries in cardiac surgery patients. Again, the discussion should be based on the results.
Author’s response:
We thank the reviewer for this comment. We have significantly revised the Discussion section to better reflect the main results of the study. The Discussion is now focused on the study’s findings, as can be seen in the following text:
Our study confirmed these findings with the addition of the fact that an elevated EuroSCORE II index certainly assumes risk for development of serious skin damages in deeper skin layers on pronounced points of the body in the early recovery period. Moreover, ejection fraction (EF) emerged as the only independent predictor of pressure injuries, indicating that patients with poorer cardiac function are at higher risk for developing PI following cardiac surgery. Although the association between reduced ejection fraction and the occurrence of respiratory, neurological, and renal complications, as well as overall mortality in cardiac patients, has been previously described 46-48, studies to date have not investigated its relationship with the development of pressure injuries following cardiac surgery. It is important to emphasize that the results showed an association between elevated EuroSCORE II index values and the development of injuries in deeper tissue layers, indicating the occurrence of stage 2 pressure injuries. Although the study by Alderden et al. 49 demonstrated that there are different mechanisms of formation, risk factors, and presentation sites for DTPI and stage 2 PI in intensive care patients, our study showed that more severe tissue injuries in cardiac surgery patients occur in areas particularly exposed to prolonged pressure, such as the heels and gluteal region. Elevated EuroSCORE II values in patients with deeper tissue damage indicate the presence of a range of intrinsic risk factors that lead to reduced tissue perfusion, which, in combination with environmental and situational factors, contributes to the development of pressure injuries. Therefore, it is important to consider parameters such as the EuroSCORE II index, which can significantly assist in identifying intrinsic risk factors with a low degree of modifiability.
Reviewer comment:
(4)The last point regarding limitations is that the authors should mention the fact that cross-sectional studies cannot provide causal relationships.
Author’s response:
We thank the reviewer for this comment. We have added the requested statement to the section of the text addressing the study’s limitations. The additional text reads as follows:
Finally, we must emphasize that the design of the study itself did not allow determination of causal relationships between the observed variables. Therefore, we recommended additional studies with a focus on the critical factors identified here and their influence on the occurrence of PI in cardiac surgery patients, as well as their impact on the overall duration of treatment and recovery after cardiac surgery.
Comments on the Quality of English Language
The English could be improved to more clearly express the research. The manuscript requires careful proofreading to address numerous language and formatting issues.
Author’s response:
We thank the reviewer for this comment. We have thoroughly revised the text of the article and hope that it now meets the requirements regarding expression in the English language.

Reviewer 2 Report
Comments and Suggestions for Authors
Comments and Suggestions for Authors
The manuscript presents a well-structured and clinically relevant study assessing the predictive role of EuroSCORE II for postoperative pressure injuries among cardiac surgery patients. The topic is timely and of interest to readers, particularly given the burden of hospital-acquired pressure injuries and the need for more precise risk-stratification tools beyond conventional existing ones.
Strengths
The introduction provides a solid contextual framework linking cardiac surgical complexity, perioperative risk, and PI pathophysiology. The study design, statistical methodology and ethical considerations are appropriate. The tables are clearly presented and support the manuscript. The findings that EuroSCORE II > 4 % is associated with a significantly higher incidence of postoperative PI are both plausible and clinically meaningful, suggesting value in integrating surgical-risk models into nursing-risk workflows.
Major Suggestions
Clarify novelty and rationale. While prior studies (Ayazoglu et al., 2018; Ettema et al., 2013) are referenced, highlight what gap this study fills or what is novel, first to validate EuroSCORE II as an independent predictor in a Central European cohort or to compare it with Braden performance.
Statistical clarity. Provide effect sizes or odds ratios quantifying the association between EuroSCORE II category and PI occurrence (p = 0.043). A multivariate logistic model adjusting for age, BMI, and operative duration may strengthen causal inference.
Terminology and consistency. Use “pressure injury (PI)” consistently throughout; some instances still use “pressure ulcer.” Define abbreviations at first mention in the text and tables.
Discussion depth. The discussion would benefit from linking results to mechanistic pathways (hemodynamic instability, tissue perfusion) and potential preventive implication (i.e. how integrating EuroSCORE II into perioperative nursing risk tools might change early mobilization, pressure-relief strategies, or ICU monitoring practices).
Limitations. If possible, please expand on potential unmeasured confounders such as nutrition, albumin, or vasopressor use, and specify whether pressure injury staging followed NPIAP 2019 criteria.
Language and flow. The English is generally clear but would benefit from minor grammatical editing for conciseness and smoother transitions (e.g., “awere” instead “were,” “statisticaly” instead of “statistically”, “date were” instead of “data was”).
Minor Suggestions
Figures or graphs visualizing PI incidence by EuroSCORE II strata may enhance readability.
Overall, the manuscript presents a sound, clinically important analysis that supports incorporating EuroSCORE II into postoperative PI risk assessment. With minor methodological clarification and editorial polishing, it would be suitable for publication.

Author Response
Major Suggestions
Reviewer comment:
Clarify novelty and rationale. While prior studies (Ayazoglu et al., 2018; Ettema et al., 2013) are referenced, highlight what gap this study fills or what is novel, first to validate EuroSCORE II as an independent predictor in a Central European cohort or to compare it with Braden performance.
Author’s response:
We thank the reviewer for this comment. We have thoroughly revised the text in the Discussion section and highlighted the important new findings of our study. The additional text relating to the novel insights that can advance current knowledge reads as follows:
„Our study confirmed these findings with the addition of the fact that an elevated EuroSCORE II index certainly assumes risk for development serious skin damages in deeper skin layers on pronounced points of the body in early recovery period. Moreover, ejection fraction (EF) emerged as the only independent predictor of pressure injuries, indicating that patients with poorer cardiac function are at higher risk for developing PI following cardiac surgery.“
We have also highlighted the aim and purpose of the study and added the main hypotheses in the Introduction section. The additional text reads as follows:
„The aim of the study is to determine the predictive value of the EuroSCORE index in assessing the risk of developing pressure injuries in cardiac surgery patients. The proposed hypotheses of the study are as follows:
H1 – Participants with a medium/high EuroSCORE II index have a higher risk of developing pressure injuries after surgery.
H2 – The occurrence of pressure injuries after surgery affects the length of the patient’s overall recovery.“
Reviewer comment:
Statistical clarity. Provide effect sizes or odds ratios quantifying the association between EuroSCORE II category and PI occurrence (p = 0.043). A multivariate logistic model adjusting for age, BMI, and operative duration may strengthen causal inference.
Author’s response:
We appreciate the reviewer's comment. To improve statistical clarity, effect sizes (Cramer’s V) were added to Table 3 to quantify the strength of the association between the EuroSCORE II risk category and the occurrence of pressure injury (P = 0.043, Cramer’s V = 0.140), indicating a small-to-moderate effect size. In addition, bivariate and multivariate logistic regression analysis was performed (Table 4). The multivariate model was adjusted for age, BMI, and duration of surgery, as suggested. In bivariate analysis, age (P = 0.002), EuroSCORE II (P = 0.001), EuroSCORE II intermediate/high risk category (P = 0.045), ejection fraction (P < 0.001), and combined surgical procedures (P = 0.022) were significantly associated with pressure injuri. After adjustment, only ejection fraction remained an independent predictor (OR = 0.97; 95% CI 0.94–0.98; P = 0.036). These additional analyses further contribute to the causal conclusions about the relationship between EuroSCORE II and pressure injuri occurrence.
Reviewer comment:
Terminology and consistency. Use “pressure injury (PI)” consistently throughout; some instances still use “pressure ulcer.” Define abbreviations at first mention in the text and tables.
Author’s response:
We thank the reviewer for this suggestion. We have thoroughly reviewed the manuscript and removed all inconsistencies in wording. Additionally, we have added the full forms of abbreviations at the points where they are first mentioned in the text.
Reviewer comment:
Discussion depth. The discussion would benefit from linking results to mechanistic pathways (hemodynamic instability, tissue perfusion) and potential preventive implication (i.e. how integrating EuroSCORE II into perioperative nursing risk tools might change early mobilization, pressure-relief strategies, or ICU monitoring practices).
Author’s response:
We appreciate the reviewer's comment. We have significantly revised the text of the Discussion section and added a series of focused links to the main findings of the study. Additionally, we have included a sentence clarifying the implications of using the EuroSCORE index in preventive nursing practice. The additional text reads as follows:
„Many studies conducted to date have clearly confirmed the association between the occurrence of pressure injuries (PIs) and prolonged ICU stay and overall hospitalization, with a significant impact on treatment costs.57-59 Although these variables are considered independent risk factors for the development of pressure injuries (PI), it is very difficult to clarify this cause-and-effect relationship. Therefore, it is important to emphasize that elevated EuroSCORE II values clearly indicate a potential risk for prolonged hospitalization, including longer surgical duration and length of stay in the ICU, which may independently represent risk factors for the development of pressure injuries. „
„Our study confirmed these findings with the addition of the fact that an elevated EuroSCORE II index certainly assumes risk for development of serious skin damages in deeper skin layers on pronounced points of the body in the early recovery period. Moreover, ejection fraction (EF) emerged as the only independent predictor of pressure injuries, indicating that patients with poorer cardiac function are at higher risk for developing PI following cardiac surgery. Although the association between reduced ejection fraction and the occurrence of respiratory, neurological, and renal complications, as well as overall mortality in cardiac patients, has been previously described 46-48, studies to date have not investigated its relationship with the development of pressure injuries following cardiac surgery. It is important to emphasize that the results showed an association between elevated EuroSCORE II index values and the development of injuries in deeper tissue layers, indicating the occurrence of stage 2 pressure injuries. Although the study by Alderden et al.49 demonstrated that there are different mechanisms of formation, risk factors, and presentation sites for DTPI and stage 2 PI in intensive care patients, our study showed that more severe tissue injuries in cardiac surgery patients occur in areas particularly exposed to prolonged pressure, such as the heels and gluteal region. Elevated EuroSCORE II values in patients with deeper tissue damage indicate the presence of a range of intrinsic risk factors that lead to reduced tissue perfusion, which, in combination with environmental and situational factors, contributes to the development of pressure injuries. Therefore, it is important to consider parameters such as the EuroSCORE II index, which can significantly assist in identifying intrinsic risk factors with a low degree of modifiability.“
„Their timely application can significantly improve nursing practice focused on preventive measures, such as implementing aids for the prevention of PI during and immediately after surgery, mobilization and monitoring of skin condition in the early postoperative period.“
Reviewer comment:
Limitations. If possible, please expand on potential unmeasured confounders such as nutrition, albumin, or vasopressor use, and specify whether pressure injury staging followed NPIAP 2019 criteria.
Author’s response:
We thank the reviewer for this suggestion. We have expanded the list of potential factors that were not investigated and included them in the study limitations. The additional text reads as follows:
„The study also did not examine the impact of other intraoperative and intrinsic factors such as hemodynamic stability, hypothermia, nutrition, albumin level and the use of vasoactive drugs.“
The classification of pressure injuries was performed according to the 2019 EPUAP guidelines, as highlighted in the Data Collection and Variables section. The text reads as follows:
„Pressure injury (PI) stage, classified according to the European Pressure Ulcer Advisory Panel (EPUAP, 2019)5 system: Stage 1 (non-blanchable erythema), Stage 2 (partial-thickness skin loss), Stage 3 (full-thickness skin loss), and Stage 4 (full-thickness tissue loss)“
Reviewer comment:
Language and flow. The English is generally clear but would benefit from minor grammatical editing for conciseness and smoother transitions (e.g., “awere” instead “were,” “statisticaly” instead of “statistically”, “date were” instead of “data was”).
Author’s response:
We thank the reviewer for this comment. We have thoroughly reviewed the text and corrected all grammatical and word pronounciation errors. The corrected text is reflected in the revised version of the manuscript.
Minor Suggestions
Figures or graphs visualizing PI incidence by EuroSCORE II strata may enhance readability.
Author’s response:
We thank the reviewer for this comment. We agree with the remark, although we believe that the tabular presentation of the results provided clear and unambiguous indicators of the objectives of the study.
Overall, the manuscript presents a sound, clinically important analysis that supports incorporating EuroSCORE II into postoperative PI risk assessment. With minor methodological clarification and editorial polishing, it would be suitable for publication.

Round 2
Reviewer 1 Report
Comments and Suggestions for Authors
Overall, the authors' revisions are satisfactory and have improved the quality of the manuscript.
However, there are a few points the author team should consider:
(1) Regarding Table 2, it is perplexing that the "z statistic" column contains two values. Primarily, the core statistic corresponding to the Mann-Whitney U test should be the U value.
(2) The authors have supplemented the results with regression analysis. It is recommended to include in the Methods section how multicollinearity was screened and addressed. This addition would be beneficial for enhancing the methodological rigor of the paper.
Comments on the Quality of English LanguageThe English could be improved.
Author Response
Reviewer’s comment:
(1) Regarding Table 2, it is perplexing that the "z statistic" column contains two values. Primarily, the core statistic corresponding to the Mann-Whitney U test should be the U value.
Author’s response:
We thank the reviewer for this comment. Each Z statistic in Table 2 corresponds to an independent Mann–Whitney U test comparing Group A (low EuroSCORE II) and Group B (medium/high EuroSCORE II) within the specified subgroup (Overall, Males, Females). To improve clarity, we have added an explanatory note below the table: “Each Z statistic corresponds to an independent Mann–Whitney U test comparing Group A and Group B within the specified subgroup.”
Although MedCalc software provides both U and standardized Z values for the Mann–Whitney test, only the Z statistic was presented in the table. The Z statistic directly reflects the standardized difference between groups and is the value used to calculate the P value, which is the common reporting convention in biomedical literature. The raw U value was therefore omitted to simplify the presentation and improve readability. A clarifying note has been added in the table legend:“Z – standardized test statistic for the Mann–Whitney U test.”. We believe this clarification adequately addresses the reviewer’s concern.
Reviewer’s comment:
(2) The authors have supplemented the results with regression analysis. It is recommended to include in the Methods section how multicollinearity was screened and addressed. This addition would be beneficial for enhancing the methodological rigor of the paper.
Author’s response:
We appreciate the reviewer's comment. To address this point, we have added a description of the screening for multicollinearity to the Statistical Analysis section. Specifically, multicollinearity among independent variables was assessed using the variance inflation factor (VIF), and all VIF values ​​were below 5, indicating no significant multicollinearity. We hope that this clarification has improved the methodological accuracy of the paper.
Comments on the Quality of English Language
The English could be improved.
Author’s response:
We thank the reviewer for this comment. We have thoroughly reviewed the text and made corrections to the grammar and language. We hope that the text now meets the linguistic requirements for publication.
